# Antioxidant Enzyme-Mimetic Activity and Neuroprotective Effects of Cerium Oxide Nanoparticles Stabilized with Various Ratios of Citric Acid and EDTA

**DOI:** 10.3390/biom9100562

**Published:** 2019-10-03

**Authors:** Ana Y. Estevez, Mallikarjunarao Ganesana, John F. Trentini, James E. Olson, Guangze Li, Yvonne O. Boateng, Jennifer M. Lipps, Sarah E. R. Yablonski, William T. Donnelly, James C. Leiter, Joseph S. Erlichman

**Affiliations:** 1Biology Department, St. Lawrence University, Canton, NY 13617, USA; boatengy@mskcc.org (Y.O.B.); jmlipps12@gmail.com (J.M.L.); seyabl15@stlawu.edu (S.E.R.Y.); jerlichman@stlawu.edu (J.S.E.); 2Psychology Department, St. Lawrence University, Canton, NY 13617, USA; 3Department of Molecular and Systems Biology, Geisel School of Medicine at Dartmouth, Hanover, NH 03755, USA; mganesana@yahoo.com (M.G.); william.donnelly@dartmouth.edu (W.T.D.); James.C.Leiter@dartmouth.edu (J.C.L.); 4Department of Emergency Medicine, Wright State University, Boonshoft School of Medicine, Dayton, OH 45435, USA; john_trentini3@yahoo.com (J.F.T.); james.olson@wright.edu (J.E.O.); guangze.li@wright.edu (G.L.); 5Department of Neuroscience, Cell Biology, and Physiology, Wright State University, Boonshoft School of Medicine, Dayton, OH 45435, USA

**Keywords:** cerium dioxide, nanoceria, ROS, catalase, superoxide dismutase, ischemia, angiotensin II, neuroprotection

## Abstract

Cerium oxide (CeO_2_) nanoparticles (CeNPs) are potent antioxidants that are being explored as potential therapies for diseases in which oxidative stress plays an important pathological role. However, both beneficial and toxic effects of CeNPs have been reported, and the method of synthesis as well as physico-chemical, biological, and environmental factors can impact the ultimate biological effects of CeNPs. In the present study, we explored the effect of different ratios of citric acid (CA) and EDTA (CA/EDTA), which are used as stabilizers during synthesis of CeNPs, on the antioxidant enzyme-mimetic and biological activity of the CeNPs. We separated the CeNPs into supernatant and pellet fractions and used commercially available enzymatic assays to measure the catalase-, superoxide dismutase (SOD)-, and oxidase-mimetic activity of each fraction. We tested the effects of these CeNPs in a mouse hippocampal brain slice model of ischemia to induce oxidative stress where the fluorescence indicator SYTOX green was used to assess cell death. Our results demonstrate that CeNPs stabilized with various ratios of CA/EDTA display different enzyme-mimetic activities. CeNPs with intermediate CA/EDTA stabilization ratios demonstrated greater neuroprotection in ischemic mouse brain slices, and the neuroprotective activity resides in the pellet fraction of the CeNPs. The neuroprotective effects of CeNPs stabilized with equal proportions of CA/EDTA (50/50) were also demonstrated in two other models of ischemia/reperfusion in mice and rats. Thus, CeNPs merit further development as a neuroprotective therapy for use in diseases associated with oxidative stress in the nervous system.

## 1. Introduction

Reactive oxygen species (ROS), including superoxide (O_2_^•−^), hydrogen peroxide (H_2_O_2_), peroxynitrite (ONOO^-^), and hydroxyl radical (HO^•^), are not only byproducts of normal cellular metabolism; they also play important roles in cellular physiology including signal transduction, adaptations to stress, and immune responses [1,2,3]. Because these molecules are highly reactive and can cause damage to proteins, membranes, and DNA, cells possess endogenous systems to neutralize excess ROS. For example, the enzyme superoxide dismutase (SOD) disproportionates O_2_^• −^ into O_2_ and H_2_O_2_ [4], whereas catalase catalyzes the conversion of H_2_O_2_ into H_2_O and O_2_ [5]. Under pathological conditions, the delicate balance between creation and destruction of ROS can be shifted, and the amount of ROS produced can exceed the capacity of endogenous antioxidant mechanisms to neutralize them, leading to a state of oxidative stress.

It is well known that oxidative stress plays a significant role in the pathology of multiple neurodegenerative conditions including stroke, amyotrophic lateral sclerosis (ALS), Parkinson’s Disease, and Alzheimer’s disease [6,7,8,9,10,11]. Edaravone is a spin trapping agent used as a drug for its antioxidant properties and was first approved for treatment of stroke in Japan. It has subsequently been approved for the treatment of early ALS in Japan and Korea, based on consistent (but not statistically significant) trends in multiple small Phase II and III studies, and later approved in the US [12,13,14]. Edaravone, like many previous antioxidant molecules, is difficult to administer and achieves limited penetration into the brain. Thus, we still lack potent and effective antioxidants that can be used therapeutically to mitigate oxidative neuronal death in pathological circumstances.

Cerium oxide (CeO_2_) nanoparticles (nanoceria or CeNPs) are potent antioxidants with a durable antioxidant effect that have recently emerged in animal models as a potential novel therapy for neurodegenerative conditions involving oxidative stress [15,16,17]. Cerium oxide nanoparticles possess both superoxide dismutase- and catalase-mimetic activity [18,19,20,21] allowing them to neutralize several important pathological ROS that can lead to neuronal injury and neurodegeneration. The ability of cerium atoms within the CeNP crystals to cycle between Ce^3+^ and Ce^4+^ valence states and the concomitant oxygen vacancies generated in the fluorite lattice structure of CeNPs permit potent, regenerative, autocatalytic activity of CeNPs [22,23,24,25]. Moreover, the small size of CeNPs allows these nanoparticles to penetrate the blood brain barrier and gain access to the central nervous system, particularly in inflammatory states in which the blood brain barrier may be more permeable [26].

While therapeutic effects of CeNPs have been reported in multiple models of neurodegeneration [26,27,28,29,30,31,32,33,34], toxic and pro-oxidant effects of CeNPs have also been described [35,36,37,38]. The physico-chemical characteristics of the nanoparticles, like nanocrystal morphology (shape, size), stabilization coatings, surface ratio of Ce^3+^/Ce^4+^, oxygen vacancies, the zeta potential [21,36,39], the biological micro-environment of the nanoparticles [40], and the adsorption of proteins and other molecules from biological fluids onto the nanoparticle surface [41] may all modulate the antioxidant activity and the ultimate biological effects of CeNPs. Thus, if CeNPs are to be developed as a treatment for neurodegenerative disorders, the physicochemical characteristics of the nanoparticles must be optimized to enhance the therapeutic effects and biological availability of the particles. Citric acid (CA), ethylenediaminetetraacetic acid (EDTA), and other small organic acids are commonly included in the synthetic reaction mixture during formation of CeNPs in order to improve nanoparticle dispersion and stability in solution [34,42,43,44], although pro-oxidant effects and poor brain penetration were observed when citric acid was the sole stabilizing agent [45,46].

We were interested in exploring how variations in the proportion of citric acid and EDTA present during synthesis modified enzyme-mimetic activity and neuroprotective effects of CeNPs. We further evaluated the neuroprotective effects of these stabilized CeNPs in two in vitro hippocampal brain slice models of ischemia in mice and in an in vivo rat model of ischemia-reperfusion.

## 2. Materials and Methods 

### 2.1. Ethics Statement

All animal experiments were performed in accordance with the guidelines set forth by the National Institutes of Health for the humane treatment of animals and the NIH Guide for the Care and Use of Laboratory Animals. The animal protocols were approved by the Institutional Animal Care and Use Committees of Dartmouth College, St. Lawrence University or Wright State University.

### 2.2. Cerium Oxide Nanoparticles 

The cerium oxide nanoparticles (CeNPs) used in this study were custom-synthesized by Cerion LLC (Rochester, NY) using a wet synthetic process [26,34]. Cerium nitrate (Ce(NO_3_)_3_·6H_2_O) was reacted with a dilute H_2_O_2_ solution at 70 °C in a high speed shearing mixer (5500 rpm) and concentrated NH_4_OH (~28–30%) was added to maintain the pH of the reaction. Various proportions (as described below) of citric acid (CA) and ethylenediaminetetraacetic acid (EDTA) were present in each reaction mixture. These organic acids functioned as stabilizers that promoted dispersion of the cerium oxide nanoparticles as they were formed in solution. Following the mixing steps, the reactor temperature was raised to 80 °C for 60 min. This mixture was cooled overnight to room temperature, then washed and filtered with a Millipore cellulose regenerated column until a pH of ~7.2 and conductivity less than 13 mS was achieved. We synthesized particles stabilized with the following ratios of CA/EDTA: 100/0; 70/30; 60/40; 50/50; 40/60; 30/70; 20/80; 0/100 (Table 1).

Particle size and composition were confirmed via X-ray diffraction (XRD) and dynamic light scattering (DLS). For XRD analysis, the stock CeNP solutions were concentrated, placed onto a zero-background quartz disk, allowed to dry under a heat lamp, and then dried in an oven for four hours at 80 °C under a vacuum. The solid sample on quartz was analyzed by XRD in a N_2_ dry cell attachment using a Rigaku D2000 diffraction system equipped with a copper anode, diffracted beam monochromator tuned to CuKα radiation, and scintillation detector. The crystallite size in the CeO_2_ (220) direction was determined using the Scherrer technique. For DLS, the CeNP stock solution was filtered (0.22 micron), and the particle size was determined using a Brookhaven DLS Instrument. Zeta potentials of the individual particle formulations were determined using a Malvern Zetasizer nano-ZS (Worcestershire, UK).

### 2.3. Enzyme-Mimetic Activity of Cerium Oxide Nanoparticles

For measuring enzyme-mimetic activity, nanoparticles were first ultra-centrifuged using an Optima™ MAX-TL Ultra centrifuge (Beckman Coulter; Brea, CA, USA) at 100,000 RPM (4.3 × 10^5^
*g*) for 1 h at 4 °C. However, the nanoceria were monodispersed, and they could not be truly pelleted and instead formed a gel-like layer of very concentrated nanoparticles that we called a “pellet” to differentiate it from the supernatant, which likely consisted mostly of dissolved cerium, citrate, and EDTA (Appendix A). Centrifugation allowed us to separate nanoparticles from diluent in order to assess the enzyme-mimetic activity of the different layers independently. Following ultracentrifugation, the layers were separately recovered using a 30-gauge insulin syringe (UltiCare; Excelsior, MN, USA).

Catalase-and oxidase-mimetic activities of CeNP fractions were measured using the Amplex Red Catalase Assay kit (ThermoFisher Scientific; Grand Island, NY, USA), as described previously [26,34]. Catalase-mimetic activity was measured using 60 µM of each CeNP fraction diluted in NaHEPES buffer (140 mM NaCl, 10 mM NaHEPES, pH 7.4). This concentration of CeNPs yielded detectable, but not saturating, levels of catalase-mimetic activity with this assay in previous studies. Oxidase-mimetic activity [47,48] may be indicative of promiscuous pro-oxidant activity that could lead to toxic effects of CeNPs. To assay oxidase-mimetic activity, we incubated higher equimolar concentrations (6.5 mM) of CeNP fractions at 37 °C for 5 h with the Amplex^®^ reagent in the absence of H_2_O_2_. The resulting fluorescence values obtained were normalized to the fluorescence measured in separate wells containing 10 µM H_2_O_2_ and the Amplex^®^ reagent incubated for the same length of time at the same temperature. Samples were assayed in triplicate and in at least three separate experiments.

SOD-mimetic activity was measured using a colorimetric SOD activity kit following the manufacturer’s protocol (Enzo Life Sciences; Farmingdale, NY, USA). A range of CeNP concentrations (6 μM–4 mM) was added to individual wells of a kit-supplied clear microtiter plate, and a mixture of xanthine oxidase and WST-1 reagent was added to each well. This was immediately followed by the addition of xanthine to initiate the production of superoxide. The superoxide produced in this reaction converts the WST-1 reagent into WST-1 formazan, which absorbs light at 450 nm. The absorbance at 450 nm was measured every minute over 10 min using a Synergy HT microplate reader (BioTek Instruments; Winooski, VT, USA). The concentration of each type of CeNP that led to a 50% inhibition (IC_50_) in the rate of absorbance change is equivalent to 1 unit of SOD activity. Samples were assayed in triplicate in at least three separate experiments.

### 2.4. Mouse Hippocampal Brain Slice Model of Ischemia

To test the neuroprotective capabilities of CeNPs stabilized with different ratios of CA/EDTA, we used a mouse brain slice model of ischemia to induce oxidative injury as described in detail previously [33,34]. For each experiment, two anatomically matched brain slices were taken from age- and sex-matched littermates: one slice was treated with the CeNPs, and the other was an untreated, ischemic control. CeNPs (5.8 μM) stabilized with different ratios of CA/EDTA or vehicle were added to the ‘test’ brain sections at the initiation of the 30 min ischemic insult, and the control brain slices were exposed to simulated ischemia without CeNP treatment. After simulated ischemic exposure, brain slices were maintained in organotypic culture for 24 h by placing them on inserts (Millipore; Billerica, MA, USA) in 35 mm culture dishes containing supplemented minimum essential culture medium (Lonza; Walkersville, MD) with CeNPs (5.8 μM; test) or vehicle alone (control). 

Twenty-four hours post-insult, paired (control and test) slices were incubated in culture medium containing 2 μM of the vital exclusion dye SYTOX Green (Invitrogen; Carlsbad, CA, USA) for 30 min. To determine cell viability, fluorescence images were captured with a Nikon TE 2000-U microscope (Nikon Instruments; Melville, NY, USA), and digital images were acquired and processed with Compix Simple PCI 6.5 software (Hamamatsu Corporation; Swickley, PA, USA). During image analysis, the total grey level was measured bilaterally in the hippocampal formation and adjacent cell layers using identical light intensity and exposure times. Two images per slice were obtained bilaterally, which was necessary to completely image the hippocampal cell layers from both sides of the brain. Total grey levels were summed for each side of the brain for each section, and the two bilateral images of the hippocampal formation were averaged to estimate the total cell death in this region. Results were expressed as the ratio of the fluorescence in the test condition (5.8 μM CeNP) to the fluorescence in the matched control (vehicle) slice images.

### 2.5. Angiotensin-II Mouse Hippocampal Brain Slice Model of Ischemia 

The renin–angiotensin system plays a critical role in hypertension, and both systemic angiotensin-II (Ang-II) and endogenous Ang-II released within the brain activate the Ang-II receptor, AT1, which leads to increased production of reactive oxygen species (ROS) by enhancement of NADPH oxidase activity [49,50,51,52,53,54,55,56]. Double transgenic mice, which over-express both human renin (R+) and human angiotensinogen (A+), develop hypertension, and this model has been a useful tool in studying the contribution of Ang-II to hypertension and oxidative brain injury. R+/A+ animals have greater tissue swelling and cell death during simulated brain ischemia compared with wild-type littermates [57]. Administration of Ang-II in wild type mice is also associated with significantly increased formation of reactive oxygen species (ROS) in brain and other tissues [58,59,60,61]. Therefore, we examined the neuroprotective effects of CeNPs during simulated ischemia in mice in the presence of exogenous Ang-II. Male C67BL mice maintained in our laboratory and aged 16 to 20 weeks were used for the experiments in this study. Brains were rapidly dissected from mice after rapid decapitation, sectioned into 400 μm slices, and the resultant slices incubated in aCSF bubbled with 95% O_2_/5% CO_2_ at room temperature for 1–2 h. After the room temperature incubation, slices were transferred to a Haas interface-type, recording stage (Harvard Apparatus; Holliston, MA, USA) and perfused for 30 min with aCSF plus 10 µM Ang-II (Sigma-Aldrich; St. Louis, MO, USA) at 35 °C under a humidified atmosphere of 95% O_2_/5% CO_2_. To simulate ischemia, the perfusion solution was changed to 35 °C glucose-free aCSF plus 10 µM Ang-II equilibrated with 95% N_2_/5% CO_2_. Simultaneous with this change in perfusion solution, the humidified gas mixture flowing over the slice was changed to 95% N_2_/5% CO_2_. Ischemic conditions were maintained for 30 min, after which reperfusion for 30 min was simulated using aCSF equilibrated with 95% O_2_/5% CO_2_ plus 10 µM Ang-II. In nanoceria-treated brain slices, CeNPs (50/50 CA/EDTA) were added at a final concentration of 5.8 µM to the perfusate during the reperfusion period.

#### 2.5.1. Measurement of ROS Production in Ang-II/Ischemia-Treated Brain Slices

Dihydroethidium (DHE) was used to evaluate ROS generation in slices during reperfusion. Non-fluorescent DHE molecules freely penetrate cell membranes where they may be oxidized to ethidium by ROS [62,63,64]. During DHE treatment, the light source illuminating the slice and other room lights were turned off to avoid photodynamic effects. After ischemia and reperfusion, slices were fixed for at least 1 h with 4% paraformaldehyde in 137 mM NaCl plus 10 mM Na_2_HPO_4_ (pH 7.4) and washed for 20 min in the same solution. The fixed slices were mounted on glass slides under coverslips using Fluoro-Gel aqueous mounting medium (Electron Microscopy Services; Hatfield, PA, USA). Photographs of the pyramidal cell layer in the CA1 region of the hippocampus and layers of the cerebral cortex immediately superior to this area were captured under epifluorescence illumination. To eliminate bias in selecting the region of interest (ROI) photographed in each brain, we placed the 10x objective over the middle of the cerebral cortex in the slice, switched the objective to 40x, focused the image without further manipulation of the stage, and acquired a single image. The number of DHE-positive cells per microscope field was counted using ImageJ software after subtracting background fluorescence. Cell density data are expressed as the number of cells per 40× microscope field.

#### 2.5.2. Assessment of Cell Death in Ang-II/Ischemia-Treated Brain Slices

Cell death was determined using propidium iodide (PI) staining of cells in the brain slices [65,66]. Following ischemia and reperfusion, slices were immediately incubated with 20 µg/mL PI in aCSF for 15 min at room temperature and then fixed overnight with 4% paraformaldehyde as described above. PI-positive cells were visualized using laser confocal microscopy, and digital images of the hippocampal formation and cerebral cortex were captured for analysis using the ROI selection procedure described above for DHE-positive cell counts. The proximal, sectioned surface of the slice was identified by adjusting the focus. To avoid the influence of cell injury at the surface caused during slice preparation, we counted only PI-positive nuclei that were situated in a single plane approximately 80–90 µm below the slice surface. The cell density of PI-positive cells was expressed as the number of cells per microscope field. 

### 2.6. Fabrication of Cyt C Microbiosensor and Electrochemical Measurements 

Biosensors were fabricated as described previously using analytical grade reagents purchased from Sigma (St. Louis, MO, USA) or Fisher Scientific (Waltham, MA, USA) [67]. Gold wires with a diameter of 0.25 mm and exposed length of 1.5 mm were modified with thiols and activated with 1-ethyl-3-(3-dimethylaminopropyl) carbodiimide (EDC) and N-hydroxy succinimide (NHS). Cytochrome c (Cyt C) from horse heart was covalently immobilized by incubating the thiol-modified gold wire electrodes in a Cyt C solution at a concentration of 5 × 10^−6^ M for 2 h. Modified electrodes were stored at 4 °C in aCSF until further use.

Electrochemical measurements, both in vitro and in vivo were performed on a computer-controlled DY2116B potentiostat (Digi-ivy, Inc.; Austin, TX, USA). Cyt C-modified electrodes were used as the working electrode, and a platinum wire was used as the counter electrode. For in vitro electrochemical measurements, a conventional Ag/AgCl electrode was used as a reference electrode (CHI Inc.; TX, USA). For in vivo electrochemical measurements, a tissue-implantable, Ag/AgCl microelectrode was used as reference electrode. The reference microelectrode was prepared by polarizing a Ag wire (0.125 mm in diameter) at +0.6 V in 0.1 M hydrochloric acid for approximately 15 min.

### 2.7. In Vivo Rat Model of Forebrain Ischemia-Reperfusion 

Adult male Sprague-Dawley rats (280-350 g; Charles River Laboratories, Wilmington, MA, USA) were housed in a temperature-controlled vivarium (21 °C) under a 12 h light:12 h dark cycle with access to standard rat chow and water ad libitum. Surgeries were performed under urethane (1200 mg/kg) and chloralose (40 mg/kg) anesthesia, and one third of the initial anesthetic dose was given as a supplement as necessary to maintain anesthesia. Body temperature was measured by a rectal thermometer and maintained at 37 °C using a heat lamp during the surgery. The surgical procedure was carried out in two steps. First, a midline cervical incision was made to expose and isolate the right common carotid artery (RCCA) from surrounding connecting tissue. An occluding cuff was placed around the exposed artery and secured in place using suture material passed through the eyelets of the vascular occluder. In the second step, the animal was fixed in a stereotaxic frame (Model 1430, David Kopf Instruments; Tujunga, CA, USA), and a midline incision was made starting caudal to the eyes and rostral to the ears to expose the surface of the skull. A small burr hole was made over the right skull to place the Cyt C biosensor in the hippocampus (stereotaxic coordinates in mm from bregma: AP: −3.8, ML: +2.0, DV: −1.5). A guide cannula was placed ca. 1 mm above the target region and fixed to the skull with dental cement. This cannula minimized damage to the sensor surface and prevented the biosensor from piercing through the hippocampus. The Ag/AgCl reference and platinum auxiliary electrodes were placed on the contralateral side in the anterior cerebral region under the skull in the CSF. The microbiosensor was implanted into the hippocampus and allowed to equilibrate for 30 min. After equilibration, superoxide levels were continuously monitored during normoxia/pre-ischemia (30 min), ischemia (15 min), and reperfusion (75 min) for a total of two hours. Ischemia was induced by inflating the occluder on the RCCA for 15 min, and reperfusion was achieved by releasing the occluder. Animals in the CeNP treatment group received 60 mg/kg of nanoceria stabilized with equal proportions of CA/EDTA (50/50) intraperitoneally 72 h before the surgery. Rats in the ischemia/reperfusion group (no CeNP administration) received an IP injection of an equal volume of vehicle (NaHEPES buffer) 72 h before RCCA occlusion, and control rats received no CeNP treatment and no ischemia. For clarity of presentation, data are shown as baseline-subtracted values. Electrochemical in vivo data were converted to concentration of superoxide in each animal based on in vitro calibrations performed for each biosensor (Appendix A). At the conclusion of each study, animals were euthanized with an excess dose of urethane and chloralose.

### 2.8. Nanoceria Content in Rat Brains

A separate group of male Sprague-Dawley rats was injected intraperitoneally with 60 mg/kg of CeNPs stabilized with equal proportions of CA/EDTA (50/50). Seventy-two hours later, each rat was euthanized by an overdose of urethane, transcardially perfused with PBS, and the brain was removed to analyze nanoceria content. Brains were frozen and analyzed by ICP-MS at the Trace Metal and Analytics facility at Dartmouth College. Brain samples (50–100 mg) were immersed in HNO_3_, heated at 105 °C for 45 min, and allowed to cool after which deionized water was added to each sample tube to achieve a final acid content of 5%. 

### 2.9. Statistical Analysis

For antioxidant enzyme-mimetic activity, a two-way ANOVA was used to compare the main effects of ultracentrifugation fraction (supernatant vs. pellet) or stabilizer ratio (CA/EDTA: 100/0; 70/30; 60/40; 50/50; 40/60; 30/70; 20/80; 0/100) and to identify any interactions between the two (fraction*stabilizer ratio; STATA 15.0). Post-hoc comparisons were made using the Bonferroni correction for multiple, preplanned comparisons. For brain slice studies assessing the neuroprotective effects of a range of CeNP formulations, paired t-tests were used to compare CeNP treatment versus vehicle-treated matched control groups. To compare the effects of CA/EDTA proportions, a one-way ANOVA was used to identify a significant main effect of the CA/EDTA proportion and a Dunnett’s post-hoc was used to compare the highest performing proportion (50/50 CA/EDTA) to the other formulations (SigmaStat version 14; Systat, San Jose, CA). For studies comparing median cell counts by treatment, a Mann–Whitney test was used (Sigma Plot version 13). For studies of ischemia and reperfusion in rats, we performed a two-way ANOVA in which ischemia/reperfusion was a repeated, within subject factor, and CeNP treatment was a between subjects factor to compare superoxide levels in vehicle-treated vs. CeNP treated rats using STATA 15.1 (StataCorp, College Station, Texas, USA).

Data are reported as the mean ± standard error of the mean (SEM) unless otherwise indicated, and *p*-values ≤ 0.05 were taken to indicate statistical significance.

## 3. Results

### 3.1. Characteristics of the CeNPs Tested 

The physico-chemical variables for CeNPs stabilized with various ratios of CA/EDTA fell within a narrow range, except for the particles at the extremes of either CA or EDTA (Table 1). Those stabilized with higher proportions of CA had higher hydrodynamic diameters, and those stabilized with higher proportions of EDTA had less negative zeta potentials. Variations in the ratio of CA/EDTA used during synthesis did not significantly alter the size of the resulting nanoparticles as determined using X-ray diffraction (XRD) in conjunction with the Scherrer equation. The crystalline structure of the CeNPs was consistent with a CeO_2_ fluorite lattice, and the size of the CeNPs measured by XRD fell within a narrow range of 2.0 to 2.5 nm (Table 1). The hydrodynamic diameter of the particles in water measured using dynamic light scattering (DLS) fell between 2.2 to 7.8 nm. The particles stabilized only with CA were the largest among the series of particles while the particles stabilized with higher proportions of CA were more variable in size according to the DLS measurements. Since the XRD measurements indicated that the sizes of the CeNP crystals were similar among all the stabilizer ratios, greater polydispersity likely represents greater agglomeration of CeNPs among those CeNPs synthesized with larger ratios of CA/EDTA. All of the zeta potentials measured were negative and ranged in magnitude from –9.1 mV to –23 mV (Table 1). Particles stabilized with higher proportions of EDTA (80% or 100%) had lower magnitude zeta potentials. 

### 3.2. Enzyme-Mimetic Activity of CeNPs 

None of the CeNPs tested precipitated out of solution upon ultracentrifugation for 1 h at high centrifugal force (4.3 × 10^5^ g), indicating that our synthetic methods generated well-dispersed and stable nanoparticles. However, ultracentrifugation generated two distinct layers: a lower layer that we called a ‘pellet,’ which was gel-like rather than solid in consistency, and a separate and distinct upper layer that we called the supernatant (Appendix A). We tested the two layers independently for enzyme-mimetic activity. 

We detected catalase-mimetic activity in both the supernatant and pellet fractions of all the CeNPs. In the supernatant, these values ranged from (mean ± SEM) 93.9 ± 26.5 mU/mL in the particles stabilized with equal proportions of CA/EDTA (50/50) to 347.3 ± 87.6 mU/mL in the particles stabilized only with CA (100/0) (Figure 1). In the pellet fractions, the measured catalase-mimetic activity ranged from 112.4 ± 17.1 mU/mL in the particles stabilized with 40/60 CA/EDTA to 504.4 ± 28.1 mU/mL in the particles stabilized with only CA. A two-way ANOVA indicated that there was a significant interaction between ultracentrifugation fraction and stabilizer ratio (F_7,74_ = 2.17, *p* < 0.05). Post-hoc comparisons indicated that the CeNPs with a CA/EDTA ratio of 100/0 had the greatest activity in the pellet and the supernatant. For CeNPs with CA/EDTA ratios less than or equal to 70/30, the catalase-mimetic activity was similar across those ratios, and similar between pellet and supernatant at each ratio.

In order to compare the oxidase-mimetic activity of nanoparticle samples across assays, we normalized the resorufin fluorescence to values obtained when the Amplex Red reagent is incubated with 10 μM H_2_O_2_ and no CeNPs at 37 °C for the same period of time (resorufin fluorescence in the presence of 6.5 mM CeNPs/resorufin fluorescence in the presence of 10 μM H_2_O_2_). Normalized oxidase-mimetic activity in the supernatant fractions was highly variable; values ranged from 0.01 ± 0.00 for the CeNPs stabilized with equal proportions of CA/EDTA to 0.37 ± 0.13 for those stabilized with 40/60 CA/EDTA (Figure 2). Normalized oxidase-mimetic activity in the pellets was generally lower than that seen in the supernatants (Figure 2). Oxidase values ranged from 0.01 ± 0.02 for CeNPs stabilized with CA only to 0.13 ± 0.05 for the particles stabilized with 40/60 CA/EDTA (F_1,58_ = 19.29, *p* < 0.0001). The denser CeNPs in the pellet displayed significantly lower oxidase activity compared to the supernatant fractions, but there was no significant interaction between ultracentrifugation fraction and stabilizer ratios in terms of oxidase activity (F_7,58_ = 1.53, p = 0.177). Interestingly, if we were to sum the average measured oxidase-mimetic activity of the pellet and supernatant fractions, the lowest combined levels of oxidase-mimetic activity were in CeNPs stabilized with equal proportions of CA/EDTA (50/50).

The SOD activity kit necessitated running a wide concentration range of CeNPs (6 μM to 4 mM) for the pellet and supernatant of each type of CeNP. Therefore, we performed assays only on the nanoparticles synthesized with CA/EDTA ratios of 100/0, 50/50, and 0/100. We detected SOD-mimetic activity in both the supernatant and pellet fractions of all the CeNPs tested (Figure 3). A two-way ANOVA indicated there was a significant interaction between ultracentrifuge fraction and stabilizer ratio (F_2,21_ = 70.30, *p* < 0.0001). Post-hoc comparisons revealed an inverse relationship between SOD-mimetic activity in the pellets compared to the supernatant. In the supernatant, CeNPs stabilized with only CA demonstrated the lowest SOD-mimetic activity (mean IC_50_ = 1265.7 ± 194.2 μM) compared with CeNPs synthesized with CA/EDTA ratios of 50/50 and 0/100. The activities of CeNPs synthesized with 50/50 and 0/100 CA/EDTA ratios were not statistically different. In contrast, SOD-mimetic activity was significantly lower in the pellets of CeNPs synthesized with 0/100 CA/EDTA compared with that measured in the pellets of CeNPs synthesized with CA/EDTA ratios of 50/50 and 100/0. Pellet SOD-mimetic activities of CeNPs synthesized with 100/0 and 50/50 CA/EDTA ratios were not significantly different (Figure 3). 

### 3.3. Neuroprotective Effects of CeNPs in a Hippocampal Brain Slice Model of Ischemia

To assess the biological effects of CeNPs stabilized with various ratios of CA/EDTA, we used a mouse hippocampal brain slice model and tested the capacity of CeNPs to provide neuroprotection after a simulated ischemic insult. Pairwise comparisons of hippocampal cell death between CeNP-treated sections and vehicle-treated control sections revealed a parabolic trend indicating that particles stabilized with equal proportions of CA/EDTA conferred the greatest protection following ischemic injury (Figure 4). Altering the relative proportions of CA/EDTA in the stabilizer from the optimum 50/50 ratio reduced viable brain tissue in this in vitro model of stroke. A one-way ANOVA using Dunnett’s test for post-hoc analyses comparing 50/50 CA/EDTA to other stabilizer combinations showed that the 50/50 stabilizer ratio was associated with significantly greater cell survival (*p* < 0.001) than seen in slices treated with the CeNPs stabilized with 100/0, 60/40, 20/80, and 0/100 ratios of CA/EDTA (Figure 4). The performance of the 50/50 CA/EDTA CeNPs was not significantly different from CeNPs synthesized with 70/30, 40/60, or 30/70 ratios of CA/EDTA (p > 0.05). Because of their potent neuroprotective effects, we subsequently tested the supernatant and pellet fractions of the CA/EDTA 50/50 CeNPs separately in this brain slice model of ischemia. We confirmed the absence of particles in the supernatant using DLS. Cell death (mean ± SEM % control) was 107.8 ± 2.3% in slices treated with only the supernatant (n = 22 paired sections) and 73.8 ± 2.9% in slices treated with only the pellet fraction (n = 16 paired sections). Thus, biological activity was associated with the CeNPs themselves resident in the pellet fractions and not some soluble factor derived from the CeNPs.

### 3.4. ROS Accumulation and Cell Death in Ang-II/Ischemia-Treated Brain Slices 

Based on the improved neuroprotection of the 50/50 CA/EDTA CeNPs in our initial brain slice ischemia model, we evaluated the performance of this material in a second, more clinically relevant model that included Ang-II. In this model of simulated ischemia/reperfusion, Ang-II (10 µM) was administered during the ischemia and reperfusion periods, and 50/50 CA/EDTA (5.8 µM) CeNPs or vehicle was present during the reperfusion period only. In Ang-II-treated brain slices exposed to CeNPs during reperfusion, ROS production was reduced 89% compared to matched hippocampal slices subjected to reperfusion without CeNPs (Figure 5A). Additionally, the number of ethidium-positive cells was significantly less in the CeNP-treated slices compared to corresponding control sections. Brain slices treated with 50/50 CA/EDTA CeNPs showed a 68% reduction in cell death compared to controls based on propidium iodide staining (Figure 5B).

### 3.5. Superoxide Monitoring in Situ in Real-Time during Forebrain Ischemia-Reperfusion

We assessed the impact of systemic treatment of rats with 50/50 CA/EDTA CeNPs in a unilateral common carotid artery occlusion model of brain ischemia and reperfusion. We made superoxide (SO) measurements in vivo in the hippocampus of anesthetized rats using a Cyt C microbiosensor in which the current response of the biosensor reflected the local tissue concentration of superoxide. A complete description of the microbiosensor, its working principle, and calibration are described in the Appendix A. The experimental timeline is illustrated in Figure 6A. The biosensor output was recorded continuously in real-time throughout the experiment. Representative real-time amperograms measured continuously in control, non-ischemic rats and rats that underwent ischemia-reperfusion and were treated either with equal volumes of vehicle or 60 mg/kg of 50/50 CA/EDTA CeNPs intraperitoneally 72 h before the ischemic insult are illustrated in Figure 6B. There was no statistically significant difference in the peak normoxic current observed between the three different treatment groups. During the ischemic period, the superoxide current started to rise and then continued to rise during reperfusion. The superoxide current also rose in CeNP-treated rats during ischemia and reperfusion, but the rate of rise of superoxide was reduced in the CeNP-treated animals compared to untreated controls. The superoxide current was stable in the control rats that experienced no ischemia and no CeNP treatment.

For quantification purposes, we defined the superoxide current responses as the difference between the current at the beginning and end of the ischemic or reperfusion period. Thus, for the ischemic period, the superoxide response was the difference in current between the end of ischemia (current at t = 45 min) and the beginning of the ischemic period (current at t = 30 min). For the reperfusion period, the superoxide current response was the difference in current between the end of reperfusion (current at t = 120 min) and the beginning of reperfusion (current at t = 45 min). The total amount of superoxide produced during ischemia and reperfusion was significantly reduced in CeNP-treated animals (0.68 ± 0.3 μM) compared to vehicle-treated controls (1.43 ± 0.7 μM). This represents an overall reduction in SO accumulation of 52% (Figure 6C) in CeNP-treated animals (F_(1,14)_ = 48.9 for the main effect of cerium treatment, *p* < 0.0001). Moreover, the formation of SO was significantly greater during reperfusion compared to ischemia (F_(1,14)_ = 15.5 for the main effect of ischemia/reperfusion; *p* = 0.0015; Figure 6C). There was no significant interaction between CeNP treatment and ischemia-reperfusion.

### 3.6. Nanoceria Content in Rat Brain

ICP-MS analysis of brain tissues of animals treated with 50/50 CA/EDTA CeNPs (single 60 mg/kg dose IP) revealed 41.4 (± 3.05) ng/g wet weight, which is consistent with the values reported in literature [26,34].

## 4. Discussion

We evaluated the antioxidant properties of CeNPs in three settings: (i) we examined the relationships among biophysical nanoparticle properties, different ratios of two anionic surface treatments used during nanoparticle synthesis, and enzyme-mimetic activity of the nanoparticles; (ii) we studied the antioxidant activity of our highest performing nanomaterial in two models of ischemia-reperfusion in brain slices; and (iii) we used a cytochrome C-based biosensor to measure the impact of CeNPs on superoxide formation during ischemia-reperfusion of the brain created by occluding right common carotid artery blood flow unilaterally. The key findings were that the different ratios of nanoparticle surface coatings were associated with different enzyme-mimetic activities and biological effects despite relatively similar biophysical properties. The CeNPs stabilized with equal ratios of CA/EDTA possessed potent antioxidant activity in all three test settings examined (in vitro, in the murine brain slices, and in vivo in the rat). While we did not focus on the tissue localization of particles in this study (i.e., intracellular/extracellular compartment), we have previously shown that CA/EDTA-coated particles gained access to the intracellular space in the biological systems tested using both anatomical (TEM) and physiological measures (quantitative fluorescence microscopy) [26,34]. We conclude, therefore, that the antioxidant and neuroprotective effects of the CA/EDTA stabilized CeNPs originate from intracellular SOD- and catalase-mimetic activity in brain tissue.

### 4.1. Particle Biophysical Properties and Antioxidant Function 

In nanoparticle form (1–100 nm scale), cerium is combined with oxygen and adopts a crystalline structure. Within the crystalline structure, oxygen vacancies form, and to maintain electronic neutrality, for every oxygen vacancy created, two Ce^4+^ atoms must be reduced to Ce^3+^ [68]. Thus, cerium oxide nanoparticles exist as CeO_2-x_, where x represents the extent of oxygen desaturation of the fluorite structure [23]. CeNPs have potent antioxidant activity—both superoxide dismutase- and catalase-mimetic activities have been described [18,20,69], and the oxygen vacancies and the low electrochemical potential barrier between the Ce^4+^ and Ce^3+^ valence states are essential mechanistic elements in the enzyme-mimetic activity. Cerium can reversibly bind oxygen and shift between oxidation states Ce^4+^ and Ce^3+^, giving the nanoparticle a regenerative capacity to act cyclically as a superoxide dismutase mimetic, which is associated with higher Ce^3+^ concentrations, and a catalase mimetic, which is associated with higher Ce^4+^ concentrations, depending on the particle size and redox environment [19,20]. Particle size plays a role in the ratio of Ce^3+^ and Ce^4+^ present in CeNPs in that as particle size decreases, Ce^3+^ content increases while Ce^4+^ declines [23,70,71,72,73]. Different analytical methods used to assess the Ce^3+^ concentration of CeNPs have yielded varying results. High vacuum TEM-derived measurements provided a value of 44% at a particle size of 3 nm [72], whereas other approaches, both experimental and mathematical, have yielded a lower estimate of the Ce^3+^ ion concentration (~25%) for 3 nm CeNPs with a 50/50 ratio of CA/EDTA [23]. 

In theory, a 50/50 ratio of Ce^3+^ to Ce^4+^ would optimally distribute the regenerative enzyme-mimetic activity between catalase and SOD activity, but smaller particles, like ours, tend to increase the bias of the particle toward Ce^3+^ valence in standardized conditions [74]. The larger ionic radius Ce^3+^ ions (101 pm in diameter) preferentially populate the surface of the CeNPs (interspersed among the Ce^4+^ ions; 87 pm in diameter), while the surface oxygen vacancies shift to just below the nanoparticle surface to reduce lattice strain [75]. There are three different types of oxygen vacancies based on the geometry of the CeNPs and the coordination number of each oxygen atom (edge, corner, and face), and different energies of vacancy formation depending upon which oxygen atoms leave the crystal to create a vacancy. The energy of vacancy formation is least at the surface of the CeNPs and greatest for oxygen molecules within the interior of the crystal. The reactivity of any particular CeNP will depend on the ease with which oxygen vacancies are formed and the ease with which the Ce^3+^/Ce^4+^ ratio can be changed, both of which depend on the location and energy state of the particular atom being changed in the crystal lattice. For example, in a 2 nm particle containing 80 Ce atoms, one half of the Ce atoms will be located on the surface (1/2 × 80 atoms = 40 Ce atoms), and the remaining half will be internal (~40 Ce ions). Using the approach described by Reed et al., if 25% of the Ce atoms are in the Ce^3+^ state based on the small size of the nanoparticle, then approximately half (20/40) of the total surface cerium atoms are in the 3^+^ state, approaching this ideal 50/50 ion balance at the nanocrystal surface where enzyme-mimetic activity takes place [23]. In addition to the impact of small nanocrystal size on the Ce^3+^/Ce^4+^ ratio, small single crystallite nanoparticles have the advantage of a large specific surface area, ~300 m^2^/gm. Thus, the combined effects of the greater representation of Ce^3+^ at the nanocrystal surface and total molar surface area of 2 nm particles provide the maximum possible number of sites for the SOD- and catalase-mimetic reactions to occur compared to other nanostructures or sizes, and cerium oxide nanocrystals around 2 nm appear to be close to an optimum size for enzyme-mimetic activity. 

While cell-free, in vitro measurements, and mathematical calculations may be useful in understanding catalytic activity under standardized conditions, their application in biological settings may be less relevant. Biological activity, which is associated with smaller particle sizes, high surface areas, and the presence of both Ce^3+^ and Ce^4+^ near parity, can vary dynamically depending on the ever-changing biological concentrations of reactive oxygen and nitrogen species [76]. Changes in cellular metabolism or injury often activate NAPDH or xanthine oxidase [77,78,79], both of which preferentially produce superoxide anion that may secondarily produce several reactive nitrogen species. Increasing the proportion of Ce atoms in the 3^+^ valence state should enhance neutralization of these ROS and enhance cellular protection against less electrophilic oxidants like O_2_^• −^, ONOO^-^, HO^•^, and nitrosoperoxycarbonate species. Thus, the rate-limiting step in redox cycling of CeNPs in biological settings may be the availability of 3^+^ sites and oxygen vacancies within the CeNPs. This is consistent with the findings of others in which the presence of a greater proportion of Ce^3+^ in the CeNP improved catalytic activity [21]. Whereas, antioxidant and anti-apoptotic effects of CeNPs were reduced in two leukocyte cell lines when Ce^3+^ levels were decreased by doping the CeNP with Sm^3+^ [80]. CeNPs have had both beneficial and toxic effects, and higher Ce^3+^ content also enhanced toxicity of CeNPs in a green alga aquatic model system [81]. The prominent effect of the valence state of CeNPs on biological activity indicates that the antioxidant and pro-oxidant effects of the CeNPs likely originate from the enzyme-mimetic activity of the nanocrystals (and not some other property of the nanoparticles). Thus, in addition to the intrinsic properties of the CeNPs that determine the availability of Ce ions in the 3^+^ and 4^+^ states, the activity of the CeNPs in biological tissue will also depend on the concentration of locally produced ROS and the cellular localization of the CeNPs. 

### 4.2. Biological Activity of CeNPs in Reduced Models

We found a parabolic relationship in neuroprotection associated with CA/EDTA proportions such that a 50/50 proportion yielded the greatest sparing in this in vitro model of ischemia-reperfusion. We confirmed both the absence of particles and the absence of biological activity in the supernatant. Moreover, the sum of enzymatic activity of the pellet and supernatant equaled the activity of the material in solution as a whole (when no centrifugation was performed), all of which indicate that the bulk of biological activity was the result of the presence of the CeNPs and not cerium or CA or EDTA ions in solution in the supernatant. In biological testing, we did not fractionate the CeNP by centrifugation, and so the biological activity in our subsequent studies is best represented by the sum of supernatant and pellet activities. 

Biological activity of CeNPs seems to be associated with access to the intracellular space where the CeNPs reduce the concentrations of a variety of ROS [33,34], although some of the neuroprotection may derive from extracellular CeNPs since most ROS are diffusible substances with half-lives sufficient to permit egress from the intracellular space where the majority of ROS generating systems are located (there are far fewer ROS generating mechanisms directed outward into the extracellular space; [78]). Consistent with our previous findings, the reduction in ROS demonstrated by the marked reduction in DHE signal (and subsequent cell death) in the mouse brain slices (Figure 5) suggests that the antioxidant effects of CeNPs were mediated by intracellular CeNPs. 

We found that the antioxidant effect of CeNPs was quite potent. Recent in vitro studies revealed a complex signaling cascade in which Ang-II, acting through the angiotensin 1 receptor (AT1R), increased intracellular Ca^2+^ release and promoted NADPH oxidase–derived formation of ROS [54,82]. Given the increased cellular, oxidative load associated with Ang-II-mediated activation of the AT1R, we speculated that elevated Ang-II levels may exacerbate neuronal cell death, particularly when coupled with an ischemic insult that elevates ROS levels independently. As shown in Figure 5, ischemia/reperfusion coupled with Ang-II administration generated significant ROS levels in hippocampal brain slices and caused significant neuronal cell death. CeNPs added only during the reperfusion period were sufficient to provide significant neuroprotection in hippocampal brain slices even when Ang-II was added to the ischemia/reperfusion exposure. Clinically, most patients receive therapy for stroke during the reperfusion period (the ischemic injury is what alerted them to the problem and occurs before therapy can be initiated). Thus, the timing of the CeNP addition during reperfusion is clinically relevant in the Ang-II/ischemia/reperfusion model. We have previously shown that nanoceria can reduce tissue death even when administered up to 3-4 h after the initial ischemic insult [33]. 

Numerous antioxidant substances have been tested in hippocampal preparations, and we found that relatively low µM concentration of CeNPs provides neuroprotection in the hippocampus in a variety of studies [33,34]. In contrast, N-acetylcysteine, which is an antioxidant frequently used to benchmark other test compounds with antioxidant activity, required a 1700-fold higher concentration (10 mM) to achieve the same level of neuroprotection provided by 5.8 μM of CeNPs in the ischemia/reperfusion model of hippocampal injury that we have used [33,34]. Edaravone, which has been used clinically in both ischemia and neurodegenerative diseases, confers neuroprotection in comparable concentrations to CeNPs in cell culture studies, but edaravone is not as potent an antioxidant. Neuroprotective effects in brain slices require 100–200 times higher concentrations of edaravone that CeNPs (5.8 μM CeNPs versus >500 μM edaravone) to achieve similar effects [83,84]. Based on the greater potency of CeNPs in these studies, CeNPs are an attractive candidate for further development as a therapeutic agent. 

In the current study, the biophysical properties did not differ that much among the CeNPs synthesized with different ratios of CA/EDTA, and it is, therefore, difficult to attribute the differences in biological activity to the biophysical properties shown in Table 1. The differences in enzyme-mimetic activity of the different CA/EDTA CeNPs that we studied cannot be based solely on Ce^3+^/Ce^4+^ valence state since the particles were similar in size and tested under identical conditions. Thus, the ratio of the anionic stabilizers must have contributed to the differences in biological function. At this time, there is no evidence that the stabilizers alter the valence state of CeNPs. The biophysical characteristics of the nanoparticles, particularly the ratio of Ce^3+/4+^ valence states measured in vitro or estimated using mathematical models, may be less relevant when the particles are in vivo and access to the intracellular space may be as important as the biophysical properties of the CeNPs and the local concentrations of the substrates for enzyme-mimetic activity may influence the balance between 3^+^ and 4^+^ states of the CeNPs as much as the intrinsic properties of the nanoparticle. Nonetheless, multiple biological studies make it abundantly clear that the surface properties of the CeNPs, however controlled, are essential determinants of the therapeutic potential of the nanoparticles [26,32,34,85].

Beyond the nanoparticle size and biophysical characteristic, other factors seem to contribute to the variation in activity associated with alterations in the proportion of CA/EDTA from the optimal 50/50 ratio. Since the major site of the antioxidant activity is intracellular, one possibility is that the CeNPs with different CA/EDTA ratios have different access to the intracellular space through either calveolin- or clathrin-mediated endocytosis, or other non-receptor mediated endocytic processes [86,87,88]. The reduction in efficacy of CeNPs that deviate further from the 50/50 CA/EDTA coating could, therefore, be associated with decreased cellular uptake. However, cellular uptake of CeNPs seems less sensitive to the surface stabilizers used since neuroprotection has been demonstrated using minimally-stabilized commercial CeNPs as well as custom CeNPs [26,33,34]. Moreover, a variety of small, uncoated nanoparticles have gained access to the intracellular space [30,89]. Therefore, it seems that the enhanced biological activity of the 50/50 CA/EDTA CeNPs is achieved by some other mechanism than increased intracellular uptake.

Much has been written about the effect of phosphate on the catalytic activity of CeNPs [90,91]. Phosphate ions poison superoxide dismutase-mimetic activity of CeNPs [73], and phosphate is present in virtually all physiological solutions in low, millimolar concentrations. Phosphate diminishes superoxide dismutase-mimetic activity by binding to the surface of the CeNP crystals at the sites of oxygen vacancies, and there is a direct relationship between the Ce^3+^ concentration and the number of oxygen vacancies, as noted above. Thus, as nanoparticle size diminishes, and oxygen vacancies at the surface increase, the nanoparticles should become more susceptible to phosphate poisoning of enzyme-mimetic activity. Given the importance of the surface interaction of phosphate with CeNPs and the adverse effect of phosphate on superoxide dismutase-mimetic activity, it seems possible that the well-preserved SOD-mimetic activity of the 50/50 CA/EDTA stabilized CeNPs derives from the negatively charged, CA/EDTA coating of the CeNPs, which may provide an electrostatic and stearic barrier reducing or preventing phosphate interactions with the oxygen vacancies at the surface of the CA/EDTA-treated CeNP. It may be that the 50/50 CA/EDTA ratio optimally reduces phosphate interactions with the oxygen vacancies of CeNP without the stabilizers themselves interfering with the superoxide interaction with the enzymatically-active, surface oxygen vacancies. A further consideration is that the stabilizers may alter the free energy of activation for enzyme-mimetic activity by altering the surface interaction between superoxide or H_2_O_2_ and the reactive sites on the CeNP surfaces to enhance the rate of catalysis independent of the Ce^3+^/Ce^4+^ valence state. Last, even a partial reduction of the interaction of phosphate with the CeNPs by CA/EDTA stabilization may be beneficial since phosphate may not reduce the catalytic activity of the CeNP sufficiently to affect the ability of the nanoparticles to perform their antioxidant functions in the environment of the cell.

Extending this argument, we may speculate that the ‘ideal’ particle for therapeutic purposes must retain both SOD-mimetic activity (Ce^3+^) and catalase-mimetic activity (Ce^4+^) when intracellular to enable cyclic, regenerative, antioxidant activity while simultaneously resisting any interactions with intracellular substances that might poison or reduce enzyme-mimetic activity. It is clear that disrupting the 3^+^/4^+^ cycling component of Ce either by doping with samarium (Sm) or by adding phosphate or by repetitive washing of the CA/EDTA particles to remove the stabilizers, as we have shown, diminishes the catalytic activity of the particles [80,91]. Thus, the Ce^3+^ valence state is a necessary but not sufficient element in biological activity. However, identifying optimal biophysical characteristics for intracellular CeNPs (or any nanoparticle) is difficult given the wide-ranging redox states and molecules a nanoparticle may encounter intracellularly over time. ‘Tuning’ CeNPs for different intracellular environments will require a detailed understanding of concentration of products, substrates, and reaction rates of endogenously produced reactive oxygen and nitrogen species as well as the intrinsic biophysical properties of the CeNPs. Currently, we have little understanding of the interactions among these processes and factors at the cellular level.

### 4.3. Biological Activity of CeNPs in Intact Animals 

This study established the usefulness of novel electrochemical Cyt C microbiosensor to monitor the dynamics of superoxide radical anion in situ in real-time during an ischemia-reperfusion injury. No information about basal levels of superoxide can be obtained with our sensor because all the data are baseline-subtracted. We explored the potential of this biosensor to estimate the relative changes in SO as a marker of tissue injury during the progression of ischemia-reperfusion injury following brief unilateral carotid artery occlusion, and we tested the hypothesis that CeNPs coated with a 50/50 ratio of CA/EDTA would reduce superoxide concentrations in this model of ischemia-reperfusion injury. We confirmed that there is excessive generation of SO during both the ischemia and the reperfusion periods (Figure 6). Moreover, CeNPs, given as a single dose three days before the right common carotid artery occlusion, effectively reduced the concentration of superoxide generated in the hippocampus during both ischemia and reperfusion. Ceria pretreatment resulted in an approximately 52% reduction of the total, combined SO accumulation during ischemia and reperfusion (Figure 6). We attribute the marked reductions in SO concentrations during ischemia and reperfusion to the superoxide dismutase mimetic activity of the CeNP. We previously demonstrated similar SOD-mimetic activity of CeNPs in a brain slice model of ischemia and reperfusion, and we were able to determine that the catalytic activity of 1 µg of nanoceria was equivalent to ~527 U of SOD activity [67]. Thus, CeNPs seem to have access to the brain, and the 50/50 CA/EDTA particle is bio-persistent and remains catalytically active, as we have shown previously [26,34]. For example, a single, 20 mg/kg injection of 50/50 CeNPs results in the retention of material for up to a year in the brain and of the catalytic activity for several weeks after delivery where they may reduce the concentration of ROS. 

The temporal dynamics of release of SO in the hippocampus in intact animals have not been reported to our knowledge. Previous studies have reported measurements of superoxide in vivo from the jugular vein [92,93], but none of them was conducted in the hippocampus, which is particularly susceptible to ischemic injury [94,95]. Moreover, SO levels in the jugular vein may reflect SO levels in the brain, but they will be the average from multiple brain regions, and the dynamics of disproportionation of SO are sufficiently fast that the SO levels in the jugular vein may bear only a distant relation to brain tissue SO levels. Thus, these studies did not provide real-time measurements of SO in the brain. There is a similar though less severe limitation in the current study. The Cyt C microbiosensor sits in an avascular pocket within the brain created during insertion; SO must diffuse over some small distance to the sensor before it can be detected. Thus, the tissue between the site of SO generation and the Cyt C biosensor acts like a low pass filter, and the SO levels measured will represent a local, smoothed, tissue averaged SO concentration.

Monitoring SO levels during disease progression in animal models would be helpful to assess oxidative stress and tissue injury in real-time so that timely interventions to reduce injury could be made. But the severity of oxidative stress can only be estimated currently based on downstream effects of ROS, such as lipid peroxidation, protein oxidation, and DNA damage [96,97]. A Cyt C biosensor can overcome the shortcomings of existing methods and facilitate real-time brain tissue measurements of SO dynamics during normoxia, ischemia, and reperfusion periods. 

### 4.4. Delivery of CeNPs to the Brain 

As a practical matter, antioxidant potency in intact animals depends on the intrinsic antioxidant power of the drug or nanoparticle, but this is confounded by access to the target organ. Thus, an intrinsically potent antioxidant may have limited biological activity if it is consumed rapidly by its antioxidant activity or if it cannot gain access to the site of ROS formation. The brain presents a particularly difficult target for antioxidant therapy since the blood brain barrier isolates the brain and limits access of most drugs. Numerous researchers have modified the surface properties of nanoparticles to increase uptake in the brain [32,85]. Particle functionalization with albumin, chitosan, and other active targeting agents has been successful in brain delivery with larger nanoparticles. Smaller, CA/EDTA-stabilized CeNPs and citrate-stabilized, non-enzyme-mimetic, metal oxide nanoparticles (<5 nm) can cross the blood brain barrier and enter the brain [26,30,34,89,98], but brain access is variable in small nanoparticles, and small differences in surface properties seem to have disproportionate effects on brain access. Independent of the size, nanoparticles rapidly accumulate protein coronas when in the circulation [41]. A protein corona may improve tissue delivery depending on the complement of proteins bound to the particle but may also increase the particle size and increase the likelihood of clearance by the reticuloendothelial system. The 50/50 CA/EDTA CeNPs bind albumin and ApoE [99], both of which can increase brain delivery through surface transcytosis or receptor-mediated transcytosis. To the extent that CA/EDTA stabilized CeNPs also possess albumin/ApoE as part of the corona, the protein corona may also facilitate entry into the brain where we have measured significant concentrations of cerium in this and previous studies. Brain deposition of ~50 ng/g tissue is sufficient to confer protection in animal models of neurodegeneration [26,34] and similar to the accumulation reported in the current study. 

Rather than rely on the vagaries of surface stabilizers or uncontrolled accumulation of serum proteins following intravenous administration, researchers have tried to design and control the surface properties of CeNPs to increase brain uptake in intact animals. PEGylation of CeNPs has been used effectively to reduce the formation of ROS and the severity of neuronal loss following cerebral ischemia in rats. Interestingly, the PEGylated CeNPs administered after the ischemic insult had better intracellular access to neurons in the site of the infarct than in the uninvolved parts of the brain [32]. Although PEGylation may have improved brain uptake, the penetration of these particles was still poor where the blood brain barrier was intact. Edaravone is an antioxidant drug approved for use in stroke and ALS, but edaravone does not cross the blood brain barrier effectively. As a consequence, the therapeutic benefit of edaravone is small at best; large (and potentially toxic) doses of the drug must be given; and edaravone is effective only in patients with early stages of ALS [100]. To overcome these limitations and to deal with the more general problem that many nanoparticles have limited access to the brain, edaravone was combined with CeNPs [85]. The core of the particle was a ~4 nm CeNP, which was surrounded by a polyethylene glycol (PEG) and angiopep-2 (ANG) coating, and edaravone was loaded into the PEG/ANG coating. The angiopep-2 was added to the outer coating to enhance transcytosis by the low-density lipoprotein receptor-related protein. These dual-coated nanoparticles gained access to the brain and reduced the intracellular concentration of ROS and reduced infarct size in an animal model of stroke. Moreover, the neuroprotective effects of CeNPs plus edaravone were greater following middle cerebral artery occlusion than the effects of CeNPs alone, demonstrating that the edaravone penetrated the intracellular space and participated, along with CeNPs, in the antioxidant effectiveness of these hybrid nanoparticles and helped reduce infarct size [85]. 

### 4.5. Limitations and Future Directions

Our studies and others aimed at correlating the properties of nanomaterials size, shape, chemical reactivity, surface charge, and composition have failed to establish a clear relationship between the physico-chemical properties of the CeNP in vitro and the biological activity of the CeNP in living tissues. It has been our hope that by analyzing different nanoparticles systematically, we would be able to determine if a single or combination of nanoparticle characteristics are responsible for specific biological effects. If this type of rational design held true, then the biological activity of a nanomaterial could be ‘tuned’ for a specific biological application a priori [101]. Despite multiple studies, the biophysical properties of CeNPs alone have not been well correlated with biological activity [81,102,103] with one exception. In nervous tissue, there does seem to be an optimal size for CeNPs that is also related to enzyme-mimetic chemistry; smaller, non-functionalized nanoparticles between 2 and 10 nm seem to confer greater neuroprotection than larger nanoparticles [22,26,28,30,34,104], and smaller CeNPs have greater Ce^3+^/Ce^4+^ ratios and a greater number of oxygen vacancies. Given the lack of correlation between physico-chemical characteristics of CeNPs and biological outcome both in the present study and others, we suggest that expanding the suite of analytical tests used to characterize nanoparticles may be helpful in determining if there are physico-chemical characteristics that correlate more closely with cellular effects.

One source of confounding information on this topic is the wide variety of preparations used by researchers, some of which are not representative of biological activity in intact animals. Unfortunately, there are no studies to date that have systematically tested the effects of varying physico-chemical properties and examined the biological outcomes in a variety of biological models (though we have been trying to accomplish this task). The dizzying array of responses to CeNPs reported in the literature, while difficult to unify, do suggest that CeNPs have biological effects (either beneficial or damaging), but the biological activity may be specific to the model system examined (i.e., cell type, immortalized cells lines or native tissue). It is apparent that small changes in the nature of the CeNP stabilizer can impact the biological effects enormously. Given this finding, comparing CeNPs between research groups becomes difficult when there are likely subtleties in both the synthesis and type of stabilizer used that significantly alter activity in living tissues. Moreover, there is a bias present in the literature that biological effects observed in one system should be conserved across cell types, tissues, and more intact preparations, an unrealistic expectation in our view. Immortalized cell lines are often used in research in place of primary, non-transformed cells or intact animals. Cell lines offer several advantages, and their use abounds in the literature examining the biological effects of CeNPs [28,36,105,106], but they have significant shortcomings as predicative, in vivo models. Both cellular function and genetics in cell lines (immortalized or primary) vary from their native counterparts [107]. Consequently, it should be understood that cell culture of either immortalized or primary cells may not fully recapitulate the physiology of native tissue, and researchers in the field should examine the activity of their particles across multiple model systems before drawing general conclusions about biological efficacy and toxicity. Once an in vitro assay/model has proven predictable in the intact animals, this method can be used as a screening tool, but only after this relationship has been established. 

## 5. Conclusions

CeNPs stabilized with 50/50 CA/EDTA show potent antioxidant enzyme-mimetic activity in vitro and also demonstrate significant neuroprotective effects in three different models of cerebral ischemia. More research is needed to establish a clear relationship between the physico-chemical properties of CeNPs in vitro and the biological activity of the CeNP in living tissues. Despite this limitation, we conclude that CeNPs merit further development as a neuroprotective therapy for use in diseases associated with oxidative stress in the nervous system. 

## Figures and Tables

**Figure 1 biomolecules-09-00562-f001:**
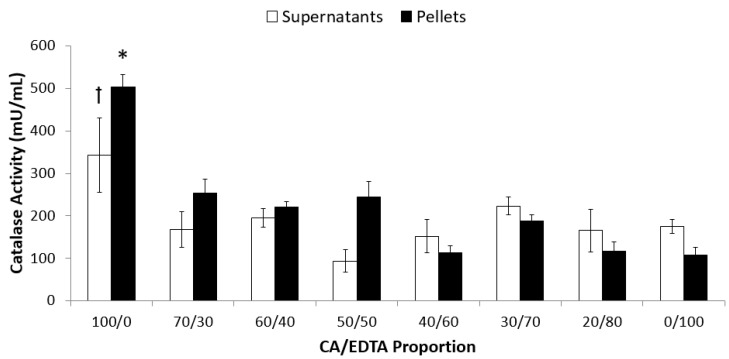
CeNPs stabilized with only CA displayed the highest catalase-mimetic activity. Catalase-mimetic activity of 60 μM of supernatant or pellet fraction of each CeNP formulation was assessed using a commercial assay kit. Data are presented as mean ± SEM of n = 4–10 separate experiments in which each formulation was assayed in triplicate. Statistical significance was determined using a two-way ANOVA and post-hoc comparisons were made using the Bonferroni correction for multiple, preplanned comparisons. Symbols indicate *p* < 0.05 compared to all other supernatant (ꝉ) or pellet (*) fractions.

**Figure 2 biomolecules-09-00562-f002:**
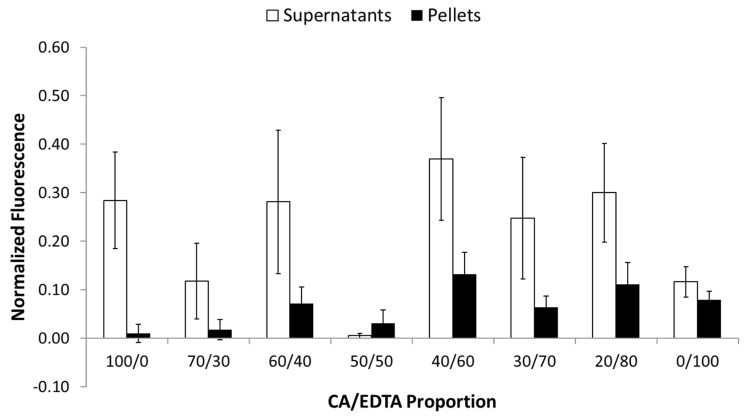
CeNPs stabilized with various ratios of CA/EDTA displayed significantly lower oxidase activity in the pellet compared to the supernatant fractions. Oxidase-mimetic activity of each CeNP formulation (6.5 mM) was assessed using a commercial kit, and data were normalized to 10 μm H_2_O_2_. Data are shown as mean ± SEM of n = 4–6 separate experiments in which each formulation was assayed in triplicate. A two-way ANOVA revealed a main effect of centrifugation fraction, indicating that lower oxidase activity existed in the pellets compared to the supernatant fractions (F_1,58_ = 19.29, *p* < 0.0001).

**Figure 3 biomolecules-09-00562-f003:**
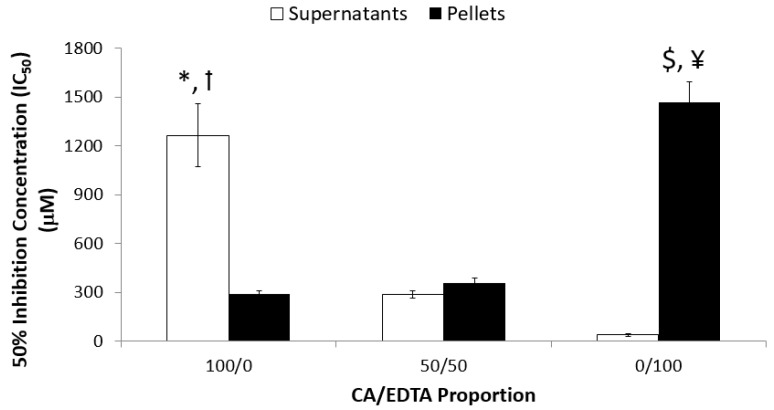
CeNPs stabilized with only CA display the highest SOD-mimetic activity in the pellet fractions, and SOD activity was highest in the supernatant fractions of CeNPs stabilized only with EDTA. SOD-mimetic activity was measured using a commercial kit (as described in the methods) where the concentration of each type of CeNP that was equivalent to 1 unit of SOD activity was determined. A higher IC_50_ indicates lower SOD activity. Data are presented as mean ± SEM of n = 4–5 separate experiments in which each sample was assayed in triplicate. A two-way ANOVA indicated an interaction between centrifugation fraction and stabilizer ratio (F_2,21_ = 70.30, *p* < 0.0001). Post-hoc comparisons were made with the Bonferroni correction for multiple, preplanned comparisons. Symbols indicate *p* < 0.05: *compared to 100/0 pellet; ꝉcompared to 50/50 and 0/100 supernatant; $ compared to 0/100 supernatant; ¥ compared 100/0 and 50/50 pellet.

**Figure 4 biomolecules-09-00562-f004:**
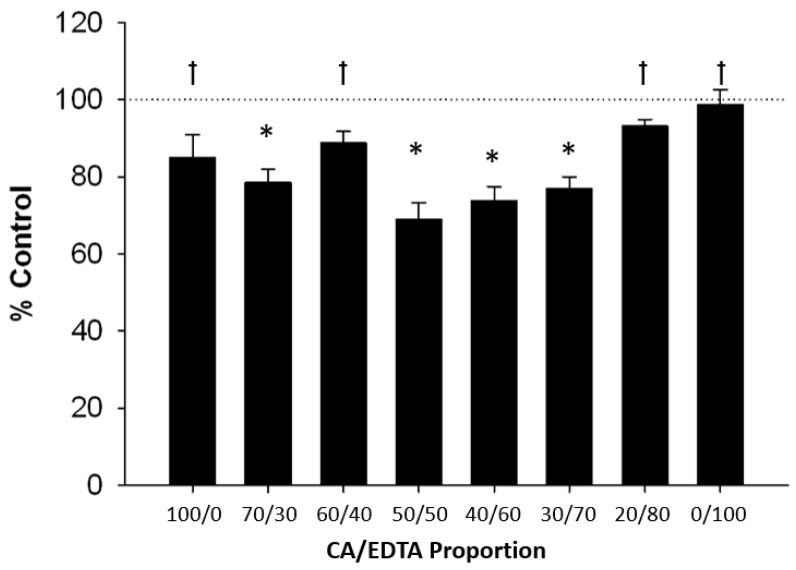
Neuroprotective activity of CeNPs stabilized with various ratios of citric acid and EDTA. Values represent the percent SYTOX fluorescence intensity compared to vehicle-treated control slices (mean ± SEM). Values less than 100% (control; dashed line) indicate reduced cell death. Statistical significance was determined by paired Student’s t-test comparing each CeNP treatment slice with a paired vehicle-treated control slice (* *p* < 0.001 compared to paired vehicle-control slice) and by a one-way ANOVA with Dunnett’s post-hoc test comparing the different ratios CA/EDTA to the 50/50 CA/EDTA († *p* < 0.001 compared to 50/50 CA/EDTA). N = 121 pairs of slices.

**Figure 5 biomolecules-09-00562-f005:**
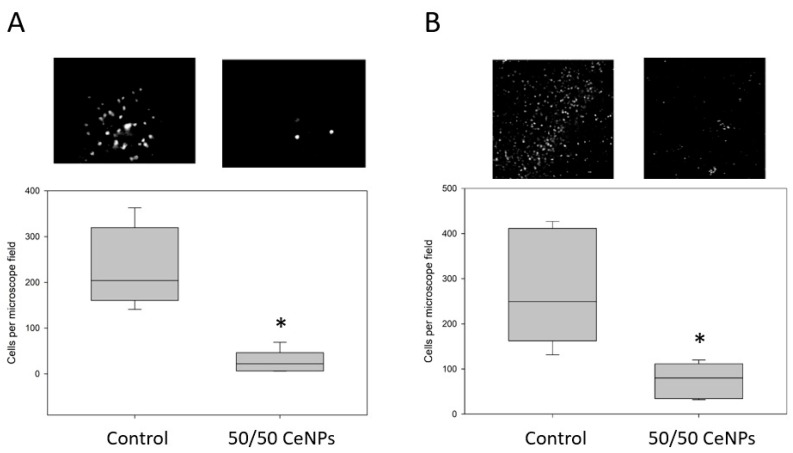
CeNPs stabilized with 50/50 CA/EDTA significantly reduce ROS (**A**) and cell death (**B**) in mouse brain slices exposed to ischemia and Ang-II as described in the methods. Values represent the median cell count per microscope field and the interquartile range. Representative fluorescence images from each group are shown above each respective box plot. Statistical significance was determined with the Mann–Whitney Rank Sum Test comparing control vs. 50/50 CeNP groups. * *p* < 0.01, n = 5–6 slices per group.

**Figure 6 biomolecules-09-00562-f006:**
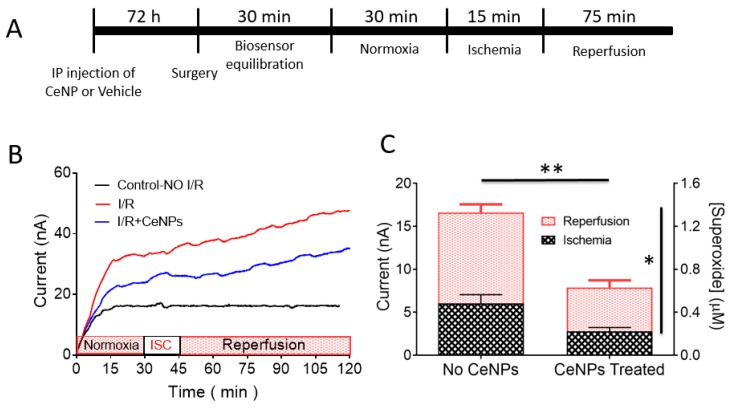
(**A**) Experimental timeline (not drawn to scale) for in vivo monitoring of superoxide during ischemia (right common carotid artery occlusion) and reperfusion in rats. (**B**) In vivo amperometric current–time response of the Cyt C microbiosensor showing continuous monitoring of superoxide anion radical in rat hippocampus at an applied biosensor potential of +0.15 V vs. Ag/AgCl. Representative amperograms for the control with no ischemia (black), ischemia and reperfusion (red) and ischemia and reperfusion in the CeNPs-treated rats (blue) are displayed. Ischemia (ISC) was induced for 15 min (30–45 min time point) and followed by reperfusion for 75 min (45–120 min time point). (**C**) Quantification of superoxide levels in vivo in rat brain hippocampus during ischemia and reperfusion. Bar graphs show the contribution of superoxide during ischemia (black hatched area) and reperfusion (red hatched area) in both CeNP-treated (n = 9) and vehicle control (no CeNPs) (n = 7) animals. Values are expressed as mean ± SEM. A two-way ANOVA revealed two main effects: a significant reduction (** *p* < 0.0001) in the total concentration of superoxide (ischemia plus reperfusion) in CeNPs-treated animals compared to controls (no CeNPs), and the SO concentration was significantly greater during reperfusion compared to ischemia (* *p* = 0.0015).

**Table 1 biomolecules-09-00562-t001:** Physiochemical characteristics of the CA/EDTA-stabilized cerium oxide nanoparticle (CeNP) series.

Citric Acid/EDTA Ratio	Size in Solution DLS (nm)	Polydispersity	Crystallite Size via XRD (nm)	Zeta Potential (mV)
100/0	7.8	0.374	2.0	−20.8
70/30	3.8	0.309	2.3	−20.4
60/40	2.6	0.198	2.4	−18.7
50/50	2.7	0.188	2.4	−21.5
40/60	2.9	0.162	2.5	−18.3
30/70	3.0	0.188	2.5	−23.0
20/80	3.5	0.160	2.4	−9.1
0/100	2.4	0.230	2.1	−15.70

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
