# Peer review of "Antioxidant Enzyme-Mimetic Activity and Neuroprotective Effects of Cerium Oxide Nanoparticles Stabilized with Various Ratios of Citric Acid and EDTA"

_biomolecules, 2019, doi:10.3390/biom9100562_

Round 1

Reviewer 1 Report

The article “Antioxidant enzyme-mimetic activity and neuroprotective effects of cerium oxide nanoparticles stabilized with various ratios of citric acid and EDTA” by Ana Y. Estevez et al. is dedicated to an experimental evaluation of biochemical properties of nanoceria, including brain slice model of ischemia experiments and in vivo testing with rat ischemia model. Authors utilized a number of truly unique methods like superoxide monitoring in vivo in the hippocampus of anesthetized rats using a Cyt C microbiosensor. The experimental results presented by authors are important, interesting, and novel.  The manuscript is of great quality, with extensive details on experimental methods and techniques used to acquire the data. In my opinion the article should be published in MDPI Biomolecules after certain minor revisions as outlined below.

Major:

The authors did not provide the explanation why the “supernatant” nanoparticles usually have higher enzyme-mimetic ability than “pellet” nanoparticles. How did they normalize the amount of applied material? Did they quantify the distribution of cerium nanoparticles among these fractions? The authors should clarify this issue.

Minor:

Line 11. Typo in the word “current” Line 96. The nitrate coefficient after parentheses need to be subscripted (Ce(NO3)3·6H2O) Line 345. A photo of the centrifuged nanoparticles would be a nice addition to supplemental materials. Line 786. Typo in the word “reticuloendothelial”

Author Response

Major:

The authors did not provide the explanation why the “supernatant” nanoparticles usually have higher enzyme-mimetic ability than “pellet” nanoparticles.

For the catalase-mimetic activity assays, there was no statistically significant difference between the enzyme-mimetic activities in the supernatant versus the pellet of each individual formulation. For the SOD-mimetic assays, we measured higher activity in the supernatant that was statistically different from the pellet only in the formulation synthesized with 100% EDTA. The formulation synthesized with 100% citric acid had greater activity in the pellet compared to the supernatant. There was no statistically significant difference between the supernatant and the pellet of the formulation synthesized with equal proportions of both CA and EDTA.

For the oxidase-mimetic assays, we did consistently observe higher oxidase activity in the supernatant compared to the pellet. Although pH can alter the oxidase activity of cerium dioxide nanoparticles (Angew Chem Int Ed Engl. 2009; 48(13): 2308–2312), we do not believe this was relevant here since the pH was similar among the different formulations.  Another potential explanation for this observation is that the oxidase-mimetic activity could be derived from dissolved cerium (Ce+3), which would be predominantly present in the supernatant as opposed to cerium nanocrystals which end up in the pellet fraction.  However, we’ve done oxidase assays with dissolved cerium (III) nitrate (unpublished observations), and detect minimal oxidase activity, eliminating this possibility. We can only speculate that perhaps breakdown products of CeNPs, or other components of the reaction such as EDTA and citric acid could have led to these results. However, this is conjecture at the moment since we have no experimental evidence to support or refute this notion.

How did they normalize the amount of applied material? Did they quantify the distribution of cerium nanoparticles among these fractions? The authors should clarify this issue.

For each experiment, we spun 2 mL of sample to make the fractions. After the centrifugation, we suspended each fraction in 2 mL final volume of buffer so that the concentration of each fraction would be the same as it was in the original sample. In addition, we were careful to handle all specimens systematically in the same way to maintain consistency across samples.

Minor:

Line 11. Typo in the word “current”.  This has been corrected.

Line 96. The nitrate coefficient after parentheses need to be subscripted (Ce(NO3)3·6H2O). This has been corrected.

Line 345. A photo of the centrifuged nanoparticles would be a nice addition to supplemental materials. Thank you for this suggestion. We added a new Supplementary Figure S1 to show an example of one of the CeNPs before and after ultracentrifugation.

Line 786. Typo in the word “reticuloendothelial.”   This has been corrected.

Reviewer 2 Report

The MS by Estevez et al. reports the “in vitro” / “in vivo” studies of neuroprotective effects of cerium oxide nanoparticles stabilized with CA/EDTA.

The article is well written, and the NPs formulation is adequately characterized. Even if the article lacks in originality (see: 1) A facile approach for synthesis of nano-CeO2 particles loaded co-polymer matrix and their colossal role for blood-brain barrier permeability in Cerebral Ischemia (2018) Journal of Photochemistry and Photobiology B: Biology, 187, pp. 184-189; 2) Role of Cerium Oxide Nanoparticles in a Paraquat-Induced Model of Oxidative Stress: Emergence of Neuroprotective Results in the Brain (2018) Journal of Molecular Neuroscience, 66 (3), pp. 420-427), I suggest to publish this paper reducing the Paragraphs 2.3/ 2.4 and Sections 3/4, paying attention only to the experimental results avoiding to describe in details all the adopted procedures recalling the previous bibliographic references

Author Response

I suggest to publish this paper reducing the Paragraphs 2.3/ 2.4 and Sections 3/4, paying attention only to the experimental results avoiding to describe in details all the adopted procedures recalling the previous bibliographic references.

Per the reviewer’s suggestion, we removed reiteration of some methodological details from the methods (paragraphs 2.3 and 2.4), the results, and the discussion sections when we already cite references that provide more details.